# Lauric Acid Induces Apoptosis of Rice Sheath Blight Disease Caused by *Rhizoctonia solani* by Affecting Fungal Fatty Acid Metabolism and Destroying the Dynamic Equilibrium of Reactive Oxygen Species

**DOI:** 10.3390/jof8020153

**Published:** 2022-02-01

**Authors:** Jian Wang, Chenglong Yang, Xianfeng Hu, Xiaolong Yao, Lei Han, Xiaomao Wu, Rongyu Li, Tingchi Wen, Li Ming

**Affiliations:** 1The Key Laboratory of Agricultural Bioengineering, Guizhou University, Guiyang 550025, China; wangj_ian@sina.com; 2Institute of Subtropical Crops, Guizhou Academy of Agricultural Sciences, Xingyi 562400, China; yangchenglong208@163.com; 3Institute of Crop Protection, Guizhou University, Guiyang 550025, China; huxianfenggzu@163.com (X.H.); 15799033071@163.com (X.Y.); lhan925@sina.com (L.H.); wuxm827@126.com (X.W.); lirongyu0328@126.com (R.L.)

**Keywords:** rice sheath blight, lauric acid, transcriptome, fatty acid metabolism, ROS imbalance, apoptosis

## Abstract

Rice sheath blight, caused by *Rhizoctonia solani*, is one of the major rice diseases. In order to better understand the inhibitory mechanism of lauric acid on the disease, RNA sequencing (RNA-Seq) was used to analyze the transcriptome changes in *Rhizoctonia solani* treated with lauric acid for 3 h, 6 h, 18 h, and 24 h, including 2306 genes; 1994 genes; 2778 genes; and 2872 genes. Based on gene ontology (GO) enrichment and Kyoto Encyclopedia of Genes and Genomes (KEGG) pathway analyses, we found that protein processing in endoplasmic reticulum (KO04141), carbon metabolism (KO01200), and starch and sucrose metabolism were significantly enriched. Most oxidoreductase, dehydrogenase, reductase, and transferase genes are downregulated in this process. Lauric acid can affect ergosterol content, mitochondrial membrane potential collapse, hydrogen peroxide content, electrolyte leakage, reactive oxygen species balance, and can induce endoplasmic reticulum (ER) stress. Lauric acid also increased the expression levels of ER chaperone glucose regulatory protein Grp78 (BIP), protein disulfide isomerase (PDI), and Calpain (CNX), and decreased the expression levels of *HSP40*, *HSP70*, and *HSP90* genes. Lauric acid affected the ergosterol content in the cell membrane of *R. solani*, which induces ER stress and increases the BiP level to induce the apoptosis of *Rhizoctonia solani*. These results indicated that lauric acid could be used to control rice sheath blight.

## 1. Introduction

Rice sheath blight, a soil-borne fungal disease [1,2], is caused by *Rhizoctonia solani* (*R**. solani*) infection, and is one of the three major diseases of rice; specifically, the second most common disease next to rice blast. The asexual state of the pathogen is *R. solani*, and the sexual state is *Thanatephorus cucumeris* [3]. It occurs globally in rice plantations in the form of hyphae or sclerotium, which reduces the yield of rice by 10–30%, and could reach up to 50% in severe disease areas [4]. The pathogen has a wide host range, and the sclerotia has tenacious vitality, reaching a germination rate of up to 12%, even after 11 years [5,6], making this pathogen difficult to control. Up to now, no immune or highly resistant varieties have been found in rice sheath blight resistance breeding, which leads to an upward trend in the incidence rate [7]. Therefore, it is very important and urgent to find effective measures to prevent and control rice sheath blight.

At present, efforts are being directed towards the development of pesticides with high efficiency, low toxicity, safety for non-target organisms, and good environmental compatibility. Plant-derived fungicides are a kind of phytoalexins that use special antibacterial substances contained in plants, and are induced to kill or inhibit the growth of target pathogenic bacteria. The fungicides are widely used, because of their high efficiency, low to no toxicity, easy degradation, high selectivity, and low drug resistance. According to reports, steroids, tannins, flavonoids, alkaloids, and saponins in plants have special antibacterial substances or induce phytoalexins to kill or inhibit microorganisms [8,9]. Yadav and Thrimurty [10] reported that mint leaf extract inhibited the mycelial growth of *Aspergillus oryzae*. Abu-Seif et al. [11] found that aromatic compounds such as phenols, flavonoids and terpenoids in lemongrass, Brazilian grass, sage, clove, and rosemary can inhibit the growth of *Aspergillus flavus*, *A. parasiticus*, and *A. ochraceus*. Studies have shown that organic compounds in lemongrass, thyme, marigold, and clove can inhibit the growth of hyphae in rice sheath blight [1,12]. Plants are called the natural treasure house of active compounds, and they produce more than 400,000 kinds of secondary metabolites, many of which have insecticidal or bacteriostatic biological activities, and have important application values in agriculture, medicine, and other fields [13,14].

Galla chinensis (GC), also known as *Aconitum carmichaelii*, Baichongcang, and Baiyao Decoction, is mainly composed of tannic acid, gallic acid, gallnut oil, lauric acid, and sugar, among others. Its main pharmacological effects are hemostasis and bacteriostasis. Zhang et al. [15] studied the effects of the water extracts of 29 Chinese herbal medicines on *Chromobacterium violaceum* (CV026) and *Agrobacterium tumefaciens* (A136), and the screening of the quorum sensing inhibitory activity of Pseudomonas aeruginosa (PAO1) showed that the water extracts of GC and Coptis chinensis could not only interfere with the production of violocin in CV026, but also inhibit the formation of biofilm in PAO1. *Galla chinensis* strongly inhibited A136 galactosidase activity, while coptis chinensis had no inhibitory effect. In the 1970s, John Kabara first proposed that glyceryl monolaurate has broad-spectrum antimicrobial activity, including against bacteria, fungi, and viruses [16,17].

It was reported in a study on the effects of fatty acids on viruses that medium-chain fatty acids (including lauric acid) and their monoglycerides were added to dairy products and proved to have significant inhibitory effects on the proliferation of influenza, vesicular stomatitis, stomatitis, and respiratory polynuclear viruses [18], mainly due to the lipid envelope and interference with cellular processes, such as transduction and transcription [19]. Desbois et al. found that the main antibacterial site of fatty acids is located in the cell membrane of bacteria, and the antibacterial effect is achieved by destroying the electron transfer system and oxidative phosphorylation on the cell membrane [20]. Richard et al. reported that the compound film of lauric acid (LDPE) can inhibit *Colletotrichum tamarilloi*, *R. solani*, and *Pythium ultimum* of wood tomato, so lauric acid could play an antifungal role in vivo and in vitro [21,22]. Among more than 40 kinds of natural fatty acids and their monoglycerides, lauric acid and glyceryl laurate were found to have better antibacterial properties [23,24,25].

Our previous research results showed that GC extract had a strong inhibitory effect on *R. solani* hyphae, while du-ensiform gall had a good toxicological effect, with a half-maximal effective concentration (EC_50_) value against the mycelia of *R. solani* of 39.23 µg/mL [26]. When we inhibited its single active component, we found that lauric acid had a strong inhibitory effect on *R. solani* hyphae. Therefore, the purpose of this study was to reveal the effect and mechanism of lauric acid on rice sheath blight, and to provide a theoretical basis for using lauric acid to control this disease.

## 2. Materials and Methods

### 2.1. Materials

*Rhizoctonia solani* was sampled in Changsha, Hunan Province. It was identified as a strong pathogenic strain (*R. solani* AG1IA) by the Institute of Crop Protection of Guizhou University. *R. solani* cultures were maintained on potato dextrose agar (PDA) at 28 °C and 150 rpm for 3 days. Some bacterial liquid was mixed with 40% glycerol (*v:v* = 1:1) and stored at −80 °C.

### 2.2. Methods

#### 2.2.1. Isolation, Culture, and Determination of Minimum Inhibitory Concentration and Lethal Dose of Rice Sheath Blight

Antifungal tests of lauric acid were carried out for assessing the effects towards the mycelial growth of *R. solani*, as described previously [26]. For the determination of contact effects, lauric acid was dispersed as an emulsion in water using 5%OP-15 (Beijing Solarbio Science & Technology Co., LTD, Beijing, China) (10% *v/v*) and added to PDA immediately before it was emptied into the glass Petri dishes (90 mm in diameter) at a temperature of 35–40 °C. The concentrations tested were 50 to 350 μg/mL. The controls received the same quantity of 5%OP-15 mixed with PDA. After solidification, a 5 mm diameter disc cut from the actively growing front of a 3-day old colony of the desired pathogenic fungus was then placed with the inoculum side down in the center of each treatment plate, aseptically. Treated petri dishes were then incubated at 28 °C till the fungal growth was almost complete in the control plates. All experiments were in quadruplet for each treatment against *R. solani*. To determine the MIC and lethal dose (LD) of lauric acid, respectively, the growth of hyphae of *R. solani* on each plate was observed after 1, 2, and 3 d of culture. The formula for calculating the growth inhibition of fungal hyphae was as follows:Inhibition (%) = (1 − Dt/Dc) × 100(1)
where Dt and Dc were the growth zone diameters in the experimental dish (mm) and the control dish (mm), respectively. Based on previous studies, the EC_50_ value was obtained by regressing growth inhibition rate against the log of the lauric acid concentration. Every treatment has three replicates.

#### 2.2.2. Scanning Electron Microscopy and Transmission Electron Microscopy Observations

PDA was poured into sterilized Petri dishes (9 cm diameter), and lauric acid was added to PDA culture mediums to make the concentrations reach 0, 50, 100, 150, and 200 µg/mL. Culture plates were incubated at 28 °C for 3 days in darkness. The hyphae were used for scanning electron microscopy (SEM) and transmission electron microscopy (TEM) observations.

Rectangular blocks (0.5 cm × 0.3 cm) from the edge of the mycelium were placed in a centrifuge tube with 1 mL of 25% dialdehyde fixation fluid. Three blocks were taken for each treatment. Each sample was suctioned repeatedly with a 50 mL syringe until the bubbles on the surface of the mycelium disappeared. The centrifuge tube was sealed and stored overnight at 4 °C. After suction, the retaining fluid was carefully rinsed three times with 0.1 M PBS for 10 min each time. Then, 0.5 mL of 1% nitric acid fixative was added within 2 h. Each sample was washed three times with PBS. Ethanol solutions of 30%, 50%, 70%, 80%, and 90% were used for dehydration for 10 min each time, followed by dehydration twice with waterless ethanol for 10 min each time. After dehydration, the specimens were dried in a freeze drier (LGJ-10D; Beijing Fourth Ring Scientific Instrument Co., Ltd., Beijing, China), and sputter-coated with gold. Microscopy was performed using an SEM (S-3400N; Hitachi, Tokyo, Japan) operated at an accelerating voltage of 20 kV. Controls consisted of untreated mycelia, which were prepared in parallel with experimental samples.

The mycelium collected as described above was poured into a centrifuge tube with 1 mL of 2.5% dialdehyde fixation fluid. The tube was sealed and incubated overnight at 4 °C. After the remaining fluid was carefully suctioned off, the sample was rinsed three times with 0.1 M PBS, for 10 min each time. Then, 0.5 mL of 1% nitric acid fixative was added within 2 h. They were then washed three times with PBS, and ethanol solutions with concentrations of 30%, 50%, 70%, 80%, and 90% were dehydrated for 10 min each time, and then dehydrated twice with waterless ethanol for 20 min each time. A mixture of acetone and resin with different concentration ratios (3:1, 1:1, 1:3 *v/v*) was successively used for infiltration for 3 h each time. This was followed by treatment with pure resin overnight. After polymerization at 70 °C for 24 h, the embedded samples were removed for the preparation of ultra-thin sections. The sections were stained with lead citrate and uranium diacetate, dried, and observed by TEM.

#### 2.2.3. Determination of the Cellular Permeability of *R. solani* to Lauric Acid

*R. solani* cake (5 mm) was taken and added to 100 mL of potato dextrose broth (PDB) medium and kept in a 28 °C shaker (150 r/h) for 24 h to form a uniform hyphal suspension. Lauric acid-incubated *R. solani* mycelia were gathered and washed twice for 2 min with sterilized water. One gram of the mycelia was placed in 15 mL centrifuge tubes containing one of three concentrations of lauric acid (0, 100, and 200 μg/mL). The background value of electrical conductivity was measured by a DDS-307 conductivity meter (J0); then after 0 h, 3 h, 6 h, 9 h, 12 h, 24 h, 48 h, according to the methods of Li et al. [27], with some modifications, 10 mL of the mixture was taken and centrifuged (4000 rpm; 5 min). Its conductivity was measured, with supernatant J1. Then, the mixture was boiled for 15 min, cooled, and centrifuged. The conductivity measured in the supernatant was recorded as J2. Finally, the relative penetration of each time period was calculated. Permeabilities (P%) were calculated by the following formula:P% = [(J1 − J0)/(J2 − J0)](2)

#### 2.2.4. Determination of Malondialdehyde Activity of Lauric Acid

The thiobarbituric acid method was used to determine the malondialdehyde (MDA) activity of lauric acid. One gram of the mycelium was weighed, and 2 mL of 10% trichloroacetic acid (TCA) and a small amount of quartz sand were added. This was ground to a homogenate, 8 mL of TCA was added, and the homogenate was centrifuged at 4000 rpm for 15 min, leaving the supernatant for later use. Two milliliters of the supernatant from centrifugation (adding 2 mL distilled water to the control) was taken and added with 2 mL of 0.6% thiobarbituric acid solution. The mixture was allowed to react in a boiling water bath for 15 min, then cooled down to room temperature. Absorbance was measured at 532 nm, 600 nm, and 450 nm (Beckman Coulter DU800 (Daniel L. Arnon), Brea, CA, USA), and the content was determined from a standard curve and calculated using the following formula:MDA (mmol·g^−1^ FW) = [6.45 × (D532 − D600) − 0.56 × D450] × [Vt/(Vs × W)](3)
where Vt: total volume of extract (mL); Vs: volume of extract for determination (mL); FW: fresh weight of extracted tissue (g). Every treatment has three replicates.

#### 2.2.5. Determination of Hyphal Ergosterol Content

The extraction method of ergosterol from *R. solani* included collecting the treated hyphae, grinding the hyphae after drying at 60 °C. Mycelium powder was weighed (0.5 g), 10.0 mL of mixed solution of methanol and chloroform (3:1) was added, and left to stand overnight at room temperature. On the next day, water, chloroform, and 0.5 mol/L phosphate buffer containing 2.0 mol/L KCl were added in turn each 10.0 mL after mixing, and extracting, and the chloroform phase was dried by a nitrogen blower at 60 °C. Ten milliliters of the mixed solution of methanol and ethanol (4:1) containing 1.4 mol/L KOH was added and saponified at 60 °C for 1 h. Ten milliliters of petroleum ether was added, extracted, and layers were separated. The petroleum ether layer was taken and blow-dried on a nitrogen blower, then dissolved with ethanol to a constant of 10.0 mL and filtered (φ Φ 0.45 µm). The lauric acid content was then determined by high performance liquid chromatography (HPLC) using an external standard method. The ergosterol standard curve was drawn according to the mass concentration. Ergosterol standard is dissolved in methanol to make stock solution and diluted into ergosterol standard (the concentration of ergosterol: 0.016, 0.032, 0.064, 0.80, 0.120 μg/mL). Then, we carry out linear regression according to the peak area and the corresponding mass concentration, and draw the standard curve.

#### 2.2.6. Transcriptome Analysis of Lauric Acid-Treated *R. solani* Mycelium

*R. solani* mycelium (1 g) was suspended in culture using PDB liquid medium containing 150 µg/mL of lauric acid and was oscillated at 150 r/min for 0 h, 3 h, 6 h, 18 h, 24 h, then centrifuged to collect the mycelium. Then, it was dried on a filter paper and freezed quickly using liquid nitrogen. Afterwards, the samples were wrapped in dry ice and sent to Beijing Baimaike Biotechnology Co., Ltd. for the next RNA extraction, Illumina library construction, and RNA-Seq sequencing. The AG-1 strains without lauric acid treatment were used as control. Three biological replicates were taken for all samples. The genome (*R. solani* AG1-IA) of *R. solani* was selected as reference sequence in this study (http://genedenovoweb.ticp.net:81/rsia/index.php?m=index&f=index, accessed on 23 July 2021).

#### 2.2.7. Bioinformatics Analysis of Sequencing Data

To control the quality of sequencing data, the raw data were first filtered to remove the sequencing connectors, sequencing primers, and low-quality clean data contained in the raw data to obtain high-quality clean data. Meanwhile, the Q20, Q30, and GC content and sequence repetition level of the clean data were calculated. All downstream analyses were based on high-quality clean data. Transcriptome assembly was completed using the Trinity software [28]. Trinity software first breaks the sequencing reads into shorter fragments (K-mer), then extends these small fragments into longer fragments (Contig), and uses the overlap between these fragments to obtain the fragment set (Component). Finally, the transcript sequence is recognized in each fragment set by using the method of De Bruijn’s graph and sequencing the read information. The functional annotation of the genes is based on the following databases: Nr (NCBI non-redundant protein sequence); Nt (NCBI non-redundant nucleic acid sequence); Pfam (protein family); KOG/COG (protein direct homologous cluster database); KO (KEGG genome-wide and metabolic pathway database); and GO (gene function annotation database). Gene expression levels were assessed using the RSEM [29] software.

#### 2.2.8. Differential Gene Expression, Gene Ontology Enrichment, and Kyoto Encyclopedia of Genes and Genomes Pathway Enrichment Analyses

The differential gene expression analysis of the control sample and lauric acid treated sample was carried out using the DESeq software (Illumina^®^; New England BioLabs, Inc., Ipswich, MA, USA). The *p*-values of the results are controlled and adjusted using Benjamini and Hochberg methods. The adjusted *p*-value < 0.05 was screened and labeled as DEG.

The enrichment analysis of the differential gene expression GO function (gene ontology) was done by the GOseq R software package [30]. Kyoto Encyclopedia of Genes and Genomes [31] is a database resource for understanding and utilizing advanced functions of biological systems such as cells, organisms, and ecosystems, and mining information from molecular level information, especially large-scale molecular datasets (http://www.genome.jp/kegg/, accessed on 23 August 2021) generated by genome sequencing and other high-throughput experimental techniques.

The statistical enrichment analysis of DEGs in the KEGG pathway was carried out using the KOBAS [32] software. To analyze whether differentially expressed genes appear (over-presentation) in a certain pathway is the pathway enrichment analysis of DEGs; the enrichment degree of the pathway was analyzed by the enrichment factor (enrichment factor) and the enrichment significance was calculated by Fisher’s exact test.

## 3. Results

### 3.1. Minimum Inhibitory Concentration and Lethal Dose of Lauric Acid against R. solani Hyphae

In the minimum inhibitory concentration (MIC) test of lauric acid, different concentrations of lauric acid showed different antibacterial effects after 2 days of culture (Table 1). Furthermore, this effect exhibited a dose-dependent effect. When the lauric acid concentration of lauric acid was less than 100 µg/mL, the plate was full of hyphae. When the lauric acid concentration was 50–200 µg/mL, a small amount of hyphae appeared on the plate. When the lauric acid concentration was greater than or equal to 250 µg/mL, no hyphae appeared on the plate. Therefore, the lowest sensitive concentration of lauric acid for the inhibition of mycelium growth was determined to be 150 µg/mL.

The experimental results of the action mode of lauric acid on *R. solani* are shown in Appendix A. When the concentrations of lauric acid were 300, 600, 800, and 1000 µg/mL for 3 days, all the bacterial cakes did not grow, but the colony diameter of the control was 7.78 cm. The bacterial cakes that did not grow were transferred to the culture medium without lauric acid. The colony diameter of the control was 6.53 cm after being cultured at 28 °C for 3 days. The mycelial growth diameters of the bacteria cakes treated with 300, 600, 800, and 1000 µg/mL of lauric acid were 5.8, 4.3, 0.85, and 0 cm, respectively. At a concentration of 1000 µg/mL, lauric acid had a lethal effect on *R. solani.*

### 3.2. Effect of Lauric Acid on the Hyphal Morphology of R. solani

The colony morphology of *R. solani* treated with lauric acid changed evidently. The hyphae of the control were uniform in thickness, and the hyphae surface was smooth, plump, and well stretched (Figure 1a,b). After a treatment of 50 µg/mL lauric acid, the hyphae showed slight wrinkles (Figure 1c,d). After a treatment of 100 µg/mL lauric acid, the hyphae had anomalies, showing the shrinkage and deformation of the hyphae and a large number of folds on the surface (Figure 1e,f). After being treated with 150 µg/mL lauric acid, the hyphae were deformed severely, with obvious folds and concave holes (Figure 1g,h). After a treatment of 200 µg/mL lauric acid, the hyphae were seriously damaged, with obvious folds, deep concave holes, and cell rupture (Figure 1i,j).

After the lauric acid treatment, the mycelial morphology of *R. solani* changed obviously, and its cell ultrastructure was severely damaged; specifically, the permeability of the mycelial cells, the structure and morphology of the cytoplasm, and the mitochondria and other organelles changed significantly. The hyphae in the control treatment had regular cell morphology and complete structure, uniform cell wall texture and thickness, a close connection with the cell membrane, dense and uniform protoplasm, and the clear and complete structure of the mitochondria, endoplasmic reticulum (ER), vacuole, and fingerprint body (Figure 2a,b). The cell wall of mycelia treated with 50 µg/mL lauric acid bent, the cells began to vacuolate, and the protoplasm gathered (Figure 2c,d). When treated with 100 µg/mL lauric acid, the cell wall shrinked obviously, the cell contents gathered, the mitochondria disintegrated, and the cell shrinkage intensified (Figure 2e,f). When treated with 100 µg/mL lauric acid, the cell wall was broken, the mitochondria in the cell were all disintegrated, showing no obvious structure, and the cell contents were aggregated (Figure 2g,h). The hyphae treated with 200 µg/mL lauric acid had serious cell wall defects, as well as presenting a complete destruction of the cell structure, and the aggregation of the cell matrix (Figure 2i,j).

In summary, the lauric acid can cause organelles in rice sheath blight cells to rupture and the mitochondria to disintegrate with an increase in lauric acid concentration, leading to physiological metabolism disorder, which indicates that on the action site of lauric acid on *R. solani* hyphae are organelles such as the intracellular mitochondria and ER.

### 3.3. Effect of Lauric Acid on the Permeability of Hyphal Cells of R. solani

The conductivity of leachate of *R. solani* hyphae treated with 100 and 200 µg/mL of lauric acid was determined by the extravasation conductivity method. The relative permeability of the mycelia to lauric acid at the beginning of the treatment made little difference. With prolonged treatment duration, the relative permeability of lauric acid-treated mycelia to floating liquid increased greatly, while the relative permeability of the blank control showed a downward trend. Then, after 3 h, the relative permeability of lauric acid-treated mycelia increased, which was higher than that of the control treatment; specifically, the relative permeability of mycelium suspension treated with 100 g/mL and 200 µg/mL of lauric acid increased by 51% and 44%, respectively, compared with that of the blank control. After 9 h of treatment, the relative permeability of mycelium suspension treated with 100 g/mL and 200 µg/mL of lauric acid increased by 31% and 66%, respectively, compared to that of the control. With prolonged treatment duration, the relative permeability of the mycelium suspension treated with lauric acid increased greatly, while the relative permeability of the blank control increased slightly after 18 h. Therefore, it was further confirmed that lauric acid treatment could improve the permeability of the cell membrane of *R. solani* hyphae and could cause some damage to the cell membrane (Figure 3). Moreover, the results showed that lauric acid treatment, at a certain concentration, could lead to damages on the hyphal cell membrane, an increase in cell membrane permeability, the leakage of electrolyte, carbohydrate, and protein, and the peroxidation of hyphal membrane lipid, and this effect was significantly enhanced with the increase in treatment concentration and duration.

### 3.4. Effects of Lauric Acid on the Malondialdehyde Content R. solani Hyphae

Malondialdehyde (MDA) is the product of membrane lipid peroxidation, and its content can reflect the degree of membrane lipid peroxidation, which is one of the important indicators of damage to the membrane system. MDA is a water-soluble small molecule, which is released into the extracellular culture medium when the cell membrane is damaged. The changes in the MDA content in the mycelia of *R. solani* after treatment with lauric acid are shown in Figure 4. After treatment with 100 and 200 µg/mL of lauric acid for 12 h, the MDA content in the mycelia and culture solution increased. The MDA content in the mycelia was higher than that of culture solution, and the difference increased with the increase in treatment concentration. After 24 h of treatment, the MDA content in the culture medium of 100 g/mL and 200 µg/mL lauric acid was higher than that of the blank control treatment, while the content in the mycelium of the untreated (CK) showed an increasing trend. It can be seen that the damage in the mycelium cell membrane caused by higher concentrations of lauric acid was significantly higher than that of the lower concentrations. Under a higher concentration of lauric acid, the MDA content in the culture medium was significantly higher than that in the hyphae, which indicated that lauric acid treatment caused cell membrane breakage, and MDA in the hyphae flowed out of the cells, and was released into the culture medium, consistent with the result that lauric acid treatment increased cell membrane permeability.

### 3.5. Effect of Lauric Acid on the Ergosterol Content of R. solani Hyphae 

Ergosterol is an important component of the fungal plasma membrane. It plays an important role in the integrity and fluidity of the cell membrane and in the maintenance of membrane protein function. The results showed that the ergosterol content of *R. solani* treated with lauric acid decreased significantly (Figure 5), which was significantly lower than that of the control. The results showed that lauric acid significantly inhibited the synthesis of ergosterol in the mycelia of *R.*
*solani* and affected the integrity of the mycelial membrane of *R. solani*.

### 3.6. Transcriptome Analysis of R. solani Treated with Lauric Acid

In order to study the molecular mechanism of lauric acid inhibiting the differential expression of rice sheath blight, the mycelia of *R. solani* were treated with lauric acid for transcriptome analysis at four time points (3, 6, 18, and 24 h). After sequencing quality control, a total of 990,713,702 raw reads were obtained, with an average of 66,047,580 raw reads and 495,356,851 clean reads per library, average clean data of 8.78 Gb, and a Q30 base percentage of over 94.05% (Table 2). The sequence generated by transcriptome sequencing was compared with the genome sequence of *R. solani* AG1-IA strain, and the comparison efficiency was over 90.16%. After assembling the transcriptome of *R. solani* AG1-IA, 13,849 transcripts and 12,569 genes were obtained. As mentioned earlier, the same database can be annotated by comparing with the public database. The 12,569 unigenes of mosaic transcripts of *R. solani* were annotated functionally in eight databases (Appendix A), and the annotation rates in the Nr and eggNOG databases in National Center for Biotechnology Information (NCBI) were 99.98% and 62.88%, respectively; in the Swiss-Prot database, the annotation rate was 42.05%; and in the Protein family (Pfam) database, it was 50.98%. According to the comparison of the annotation results (Appendix A) with the Nr database, the AG1-IA strain with the highest homology with rice sheath blight had an annotation rate of 99.98%.

### 3.7. Identification of Differentially Expressed Genes (DEGs)


In order to analyze the differences in the gene expression of *R. solani* treated with lauric acid for different time points, and to compare the changing rules of gene expression after 3, 6, 18, and 24 h of treatment with lauric acid, differentially expressed genes (DEGs) were identified. After treatment with lauric acid for 3 h, *R. solani* was induced by lauric acid to express 2306 differential genes, including 1142 upregulated genes and 1164 down-regulated genes, as compared with the control. After 6 h of treatment with lauric acid, 1994 differentially expressed genes were identified in *R. solani*, including 850 upregulated genes and 1144 down-regulated genes. There were 2778 differentially expressed genes in R. solani after 18 h of treatment with lauric acid, including 1246 upregulated genes and 1532 down-regulated genes. After 24 h of treatment with lauric acid, there were 2872 DEGs, including 1284 upregulated genes and 1588 downregulated genes in *R. solani* (Figure 6a). According to the differential gene analysis of *R. solani* treated with lauric acid at different time points compared with the control, there were 742 differential genes unique to the treatment after 3 h; 263 genes with unique differences after 6 h; and 186 genes with unique differences after 18 h. There were 316 genes with unique differences after treatment at 24 h. The total number of DEGs at all time points was 781. We used hierarchical clustering for all the DEGs to observe the gene expression patterns, and the expression levels of each gene were calculated using the fragments per kilobase million mapped reads (FPKM) method (Figure 6b).

### 3.8. Gene Ontology and Kyoto Encyclopedia of Genes and Genomes Analyses of Enriched Differentially Expressed Genes after Different Treatments

In order to study the function of the gene expression profile, the identified genes were analyzed by gene function and enriched by the Kyoto Encyclopedia of Genes and Genomes (KEGG) pathway. We did a gene ontology (GO) analysis for three main categories: biological processes, cellular components, and molecular functions. We found that genes related to fatty acid synthesis and metabolism are shared; that is, the genes were participating in lipid transport and metabolism, carbohydrate transport and metabolism, cell wall/membrane/envelope biogenesis, and defense mechanisms, among others. These results indicated that lauric acid inhibited the growth of *R. solani* hyphae and mainly destroyed the structure of the cell membrane. After an analysis of the differential gene expression of *R. solani* after lauric acid treatment (Appendix A, Appendix A) and the GO enrichment analysis of DEGs, which participated in 40 significant functional GO classifications, the significantly enriched categories of the biological process were metallic process, cellular process, single-organic regulation, and localization. In the category of “cell components,” the most important category determined was firstly, “membrane, membrane part, cell, cell part” and secondly, “organelle, macromolecular complex, organelle part, membrane-enclosed lumen, extracellular region”. In addition, among the “molecular function” categories, the GO function category with the largest number of DEGs was “catalytic activity,” followed by “binding” and “transporter activity”.

In order to identify the GO categories significantly enriched in DEGs, the GO enrichment analysis was carried out with the agriGO online tool (*p* ≤ 0.05), taking all GO items of unigenes as reference. The analysis results showed that among these DEGs, there were 40 significantly enriched GO categories, including 15 “cell component” categories, 12 “molecular function” categories, and 13 “biochemical process” categories. However, the 10 GO categories had the most significant enrichment degree in each category.

After the lauric acid treatment for 3, 6, 18, and 24 h, KEGG a pathway biosynthesis of antibiotics (ko01130), protein processing in endophytic retinium (ko04141), carbon metabolism (ko01200), starch and sucrose metabolism (ko00500), were the common pathways that were significantly enriched. The enrichment analysis of metabolic pathway genes that different genes may participate in via KEGG enrichment is shown in Appendix A. After lauric acid treatment, *R. solani* caused a series of differential gene expressions of metabolic or synthetic pathways related to energy or secondary substances (Appendix A). The main pathways were: protein processing in ER (ko04141), biosynthesis of antibiotics (ko01130), biosynthesis of amino acids (ko01230), carbon metabolism (ko01200), and starch and sucrose metabolism (ko00500), among others. In addition, with the prolongation of lauric acid treatment time (3, 6, 18, and 24 h), the number of DEGs involved in the metabolic pathway gradually increased, which were 504, 409, 545, and 580, respectively. Generally speaking, in the early stage of lauric acid treatment, *R. solani* responded to lauric acid, mainly participating in metabolic and energy-related pathways. With the prolonged duration of lauric acid treatment, the number of DEGs enriched by participating pathways increased, which made *R. solani* resist the inhibition of lauric acid. In addition, after lauric acid treatment for 3, 6, 18, and 24 h, the gene expression in the enrichment pathway was significantly enriched by ribosome, pyrimidine, and glutathione metabolism after 3 h and 6 h, which indicated that the response of *R. solani* to lauric acid was related to cell function. After treatment for 18 h and 24 h, the main pathways were the secondary metabolic pathway, fatty acid metabolism, protein processing, peroxisome, and so on, which could induce cell apoptosis and inhibit the hyphal growth of *R. solani*. Therefore, it is of great significance to explore the differential genes involved in the above metabolic pathways for a deeper analysis of the inhibiting mechanism of lauric acid on *R. solani*.

### 3.9. Analysis of Starch and Sucrose Metabolism Pathways in R. solani Induced by Lauric Acid

Thirty-seven DEGs of *R. solani* induced by lauric acid were related to starch and sucrose metabolism. Lauric acid regulates the genes for the key enzymes in starch and sucrose metabolism, upregulates the expression of the β-fructofuranosidase (EC: 3.2.1.26) gene, and converts UDP-D-glucose into β-D-fructose. In addition, β-d-glucosidase (EC: 3.2.1.21) and α-amylase (EC: 3.2.1.1), which are involved in the mutual transformation between cellulose and cellobiose, starch and dextrin, maltodextrin and maltose, trehalose and glucose, fructose phosphorylation, glucose phosphorylation, and other enzymes related to the decomposition of polysaccharides into glucose were all downregulated (Figure 7).

### 3.10. Effect of Lauric Acid on the Fatty Acid Metabolism of R. solani

The fatty acid metabolism of *R. solani* is activated in the peroxisome, and acyl-CoA synthase (EC: 6.2.1.3; ACSs) catalyzes the production of fatty acyl coenzyme A, which has high-energy bonds in its molecule and is active in nature. Peroxisomal biogenesis factor 5 (PEX5) combines with peroxisomal targeting signal type 1 (PTS1) and enters the peroxisome; then, the long-chain acyl-CoA synthetase (EC: 6.2.1.3; LACS) gene is upregulated to catalyze the generation of hexadecyl coenzyme A. Acyl-CoA oxidase (EC: 1.3.3.6) in lauric acid-treated *R. solani* (ACDs) gene is upregulated with fatty acyl-CoA dehydrogenase (acyl-CoA dehydrogenase, EC: 1.3.8.8; ACOX) and produces trans-hexadecyl-2-enoyl coenzyme A, which is reacted with enoyl-CoA hydratase (enoyl-CoA hydratase, EC: 4.2.1.17; ECHDC) and is further catalyzed to (S)-3-hydroxy-hexadecyl-coenzyme A. The gene for 3-hydroxyacyl-coenzyme A dehydrogenase (3-hydroxyacyl-coa dehydrogenase, EC: 1.1.1.35; HOAD) is upregulated, and then reacted with long chain hydroxyacyl-CoA dehydrogenase long-chain 3-hydroxyacyl-CoA dehydrogenase, EC: 1.1.1.211; LCHAD) to produce 3-oxoisopentenyl coenzyme A, which is catalyzed by acetyl-CoA acyltransferase (EC: 2.3.1.16; ACAA). This catalyzes the production of coenzyme A and tetradecanoyl coenzyme A. After β-oxidation, the carbon chain of fatty acid is shortened, then acyl coenzyme A and medium/short chain acyl coenzyme A enter mitochondria and undergo β-oxidation, finally forming acetyl coenzyme A, which provides energy or other ways for the substrate to enter the body through the tricarboxylic acid (TCA) cycle (Figure 8). Under lauric acid stress, in order to resist the poison of lauric acid, the fatty acid metabolism and degradation process in the pathogen produced excessive oxygen free radicals, changed the membrane structure, and oxidized the cell membrane, which led to the destruction of the membrane structure of mitochondria or ER, finally inducing cell apoptosis.

### 3.11. Effects of Lauric Acid on the Antibiotic Biosynthesis and Carbon Metabolism of R. solani

When faced with environmental stress, fungi actively change their metabolic and synthetic pathways to adapt to changing environmental conditions. After treating *R. solani* with lauric acid for 3 h and 6 h, related genes such as isoprene biosynthesis-related protein, pyruvate dehydrogenase, acetolactate synthase, citrate synthase, aspartate amino-lyase, refined amino succinic acid synthase, malate dehydrogenase, and branched chain amino acid transaminase were upregulated, which participated in the synthesis of terpenoids, amino acid synthesis, glycosides and biomacromolecules in energy metabolism, and increased the drug resistance of *R. solani* (Figure 9a). After lauric acid treatment of *R. solani* for 18 h and 24 h, the related enzyme genes in the TCA cycle pathway, sugar degradation pathway, pentose phosphate pathway, and arginine synthesis pathway were downregulated, and the substrate and energy in antibiotic biosynthesis were deficient, resulting in decreased antibiotic synthesis speed and drug resistance of *R. solani*. It is speculated that the pathogen of rice sheath blight can upregulate amino acid metabolism, sugar metabolism, and oxidative phosphorylation to synthesize antibiotics through a signal transduction pathway in the early stage of lauric acid treatment, and enhance the effect of bacteria on lauric acid. With the extension of stress time, the stress degree of *R. solani* was aggravated, and the stress environment inhibited its metabolic activity by downregulating the expression of more genes, thus causing some damage to *R. solani*.

Under lauric acid treatment, the pathogen of rice sheath blight caused a change in the transcription level of carbon metabolism-related genes, which provided raw materials and energy for it to resist the stress of lauric acid (Figure 9b). Ethanol dehydrogenase (EC 1.1.1.1) caused by lauric acid treatment of *R. solani* for 3 h (ADH), formate dehydrogenase (EC: 1.2.1.2; FDH), malate dehydrogenase (EC 1.1.1.37; MDH), malate synthase (EC2.3.3.9; MS), citrate synthase, (EC2.3.3.1; CS), and other related genes were upregulated. After the lauric acid treatment of *R. solani* for 6, 18, and 24 h, glucose-6-phosphate isomerase (EC: 5.3.1.9; GPI), NADP^+^ glyceraldehyde-3-phosphate dehydrogenase (EC: 1.2.1.9; GPD), phosphoenolpyruvate carboxylase (EC: 4.1.1.31; PPC), succinate dehydrogenase (EC: 1.3.5.1; SD), hexokinase (EC: 2.7.1.1; HK), D-3-phosphoglycerate dehydrogenase (EC: 1.1.1.95; PGD), acetyl-CoA C-acetyltransferase (EC: 2.3.1.9; AACT), and glyoxylate/succinic semialdehyde reductase (EC: 1.1.1.79; G/SSR) genes were downregulated. When the pathogen of rice sheath blight was stressed by lauric acid, the pathogen gradually released energy through catabolism and intracellular respiration to maintain the supply of energy and raw materials. With the increase in stress time, the genes of sugar degradation, lipid metabolism, TCA cycle, and other related enzymes of *R. solani* were downregulated, which led to the excessive accumulation of electrons in cells combining with oxygen to form superoxide, destroying the integrity of the cell membrane system and leading to cell rupture.

### 3.12. Effect of Lauric Acid on the Endoplasmic Reticulum Protein Processing Pathway of Rhizoctonia solani

Forty-five DEGs of *R. solani* induced by lauric acid were related to the ER protein processing pathway. In the ER protein processing pathway, the calcium calnexin (CNX) gene and glucose regulatory protein 94 gene (*GRP94*) were upregulated, which indicated that ER protein processing mainly focused on the degradation pathway (ERAD), and differentially expressed genes which mainly participated in three processes of substrate recognition, transport and degradation by proteasome after ubiquitination in the ERAD pathway (Figure 10). *Derlin*, which promotes the release of substrate during transportation, upregulated its expression, and transmits the protein to the related enzyme gene *OS-9* and *XTP3-B* of the ubiquitinase complex. The heat shock protein (HSP) protein family can bind unfolded or misfolded proteins and promote their correct folding. The *HSP40*, *HSP70*, and *HSP90* genes were all downregulated; thus, proteins could not fold correctly. Among the DEGs in the ubiquitinase complex components, the core component of the HRD1/HRD3 ubiquitinase complex, the *HRD1* ubiquitin ligase gene was upregulated, while the E2 ubiquitin binding enzyme gene *Skp1* was downregulated, so that the ubiquitinase complex cannot be formed, and the substrate protein cannot be ubiquitinated. These results indicated that the ER-associated degradation (ERAD) system of *R. solani* treated with lauric acid could not accurately degrade unfolded or misfolded proteins, thus inducing the apoptosis of *R. solani* cells.

## 4. Discussion

Rice sheath blight is one of the diseases that have a great economic impact on rice production. At present, the main measures to control rice sheath blight are agronomic management, chemical control and biological control, and fungicides, with fungicides determined as the most effective means to control rice sheath blight so far [6,33]. Jinggangmycin is the most commonly used fungicide to control rice sheath blight in rice plantations in China, and its high dependence may lead to many problems, such as drug resistance, pesticide residues, and excessive dosage, among others. Long-term use will cause immeasurable harm to human beings and the environment [34]. Biological control instead of chemical control is undoubtedly one of the most effective means for rice sheath blight. Fungal and bacterial biological control agents have been used to control rice diseases such as *Trichoderma harzianum*, *Bacillus subtilis*, *Pseudomonas*, and *Streptomyces* [35,36]. Combined with antibiotics to achieve the best prevention and control of rice sheath blight, the above measures have significant advantages in protecting environment safety, and there are many shortcomings compared with traditional chemical control. The unstable effect of biological control is one of the main problems in practical application, and the lack of compatibility between complex environmental conditions and other pesticides is considered as the main reason for its instability. Therefore, it is necessary to identify alternatives such as plant extracts, biological agents, and a new generation of fungicides for comprehensive use in the outbreak of rice sheath blight.

Rice sheath blight is a major disease that seriously affects rice yield. However, the mechanism of lauric acid defense against this disease is still unknown. In this study, we conducted an RNA-Seq-based transcriptome analysis to analyze the molecular mechanism of lauric acid in the inhibition of the mycelia of *R. solani*. Under these conditions, after *R. solani* was treated with lauric acid for 3 h, compared with the control, *R. solani* was induced to express 2306 DEGs, including 1142 upregulated genes and 1164 downregulated genes. After 6 h of treatment with lauric acid, there were 1994 DEGs in *R. solani*, of which 850 were upregulated, and 1144 were downregulated. After 18 h of lauric acid treatment, there were 2778 DEGs in *R. solani*, of which 1246 were upregulated, and 1532 were downregulated. After 24 h of lauric acid treatment, there were 2872 DEGs, of which 1284 were upregulated, and 1588 were downregulated. These genes were further enriched and analyzed by GO analysis and were found to participate in 40 significant functional GO classifications, among which the significantly enriched biological processes were the metallic process, cellular process, single-organization regulation, and localization. In addition, the KEGG analysis showed that the DEGs were divided into the biosynthesis of antibiotics (ko01130), protein processing in ER (ko04141), carbon metabolism (ko01200), and starch and sucrose metabolism (ko00500). The GO functional enrichment showed that these genes were involved in cell membrane integrity, metabolic synthesis, and redox process, among others, and that there were positive correlations with active oxygen homeostasis. The KEGG pathway analysis showed that DEGs involved in the biosynthesis of secondary metabolites, including amino acid synthesis, antibiotic biosynthesis, fatty acid metabolism, and pyrimidine metabolism were downregulated in lauric acid-treated *R. solani*. In response to environmental stress, cells need to synthesize more secondary biomass, so they need to consume more energy. At the same time, the energy demand for repair and synthesis also increases. Therefore, cells need to choose the most effective way to generate more energy and maintain energy reserves under environmental stress. In addition, the MAPK signal transduction pathway is also activated, and plays an important role in hyphal growth regulation [37]. Our results show that treating *R. solani* with low concentrations of lauric acid for 3 h cannot significantly inhibit rice sheath blight, but the apoptosis of hyphae is related to the concentration and treatment time of lauric acid. Specifically, *R. solani* destroys the integrity of the hyphal membrane and inhibits the growth of hyphae through MAPK interaction, active oxygen accumulation, and lipid metabolism; thus, achieving the purpose of preventing the growth of hyphae after treatment with lauric acid.

Reactive oxygen species (ROS; e.g., H_2_O_2_) can damage many components in cells, and cells are under oxidative stress, which leads to lipid peroxidation, protein oxidation, nucleic acid damage, and enzyme inactivation, and can activate programmed cell death [38]. Reactive oxygen species accumulation is related to plant immune response [39], and immune response is related to disease resistance. We found that lauric acid can oxidize the membrane system, destroy the integrity of the cell membrane, and enhance the permeability of the membrane in *R. solani*, thus affecting mycelium growth. Plant polygalacturonase inhibitor proteins (PGIPs), tobacco NaD1, Hcm1 protein, and pectin lyase induced the H_2_O_2_ accumulation of cotton verticillium wilt pathogen, and cell membrane oxidation destroyed the cellular structure and induced cotton disease resistance [40,41,42,43]. Hydrogen peroxide can oxidize the sulfur-containing groups of cysteine and methionine and inactivate some enzymes (such as the sugar enzymes fructose-1,6-diphosphatase, hexokinase, succinate dehydrogenase, and ribulose phosphate kinase) in the peroxidase cycle, TCA cycle, and sugar degradation. When treated with lauric acid for 18 h and 24 h, the genes encoding these metabolic pathway-related enzymes were downregulated, resulting in a large amount of intracellular H_2_O_2_ accumulation, and the oxidation of protein kinase, phosphatase, and transcription factors containing thiol residues. Lauric acid can inhibit hyphal growth and destroy the cellular membrane system and has no adverse effect on rice seedlings; thus, it can be used to control rice sheath blight. Moreover, lauric acid has no adverse effects on rice growth, and is also suitable for a wide range of temperature and pH levels; thus, it can be directly applied to rice plants as a fungicide to control rice sheath blight. There are few reports that lauric acid can inhibit *R. solani* or other pathogens, which indicates that lauric acid has a good effect on preventing and treating *R. solani*. We speculated that lauric acid could induce the apoptosis of *R. solani* and improve the control effect of rice sheath blight.

We present a mechanism model of the cell apoptosis induced by lauric acid in *R. solani*. Our results showed that lauric acid induced the upregulation of glucanase and fatty acid metabolism-related genes, increased the amount of ergosterol in the cell membrane, led to cell membrane damage, and downregulated cytochrome oxidase and upregulated iron reductase, resulting in the accumulation of intracellular ROS (H_2_O_2_). The excessive production of ROS leads to cell membrane potential collapse, membrane oxidation, enhanced membrane permeability, and electrolyte exosmosis, resulting in cell membrane destruction. Moreover, ROS accumulation causes DNA fragmentation and ER stress, activates phosphatases, and causes the apoptosis of *R. solani* cells. These are the main mechanisms by which lauric acid destroys the rice sheath blight pathogen (Figure 11).

Plant organic compounds have strong hydrophobicity to destroy the integrity of the cell membrane, which leads to infiltration and thereby inhibits cell growth. Cell membrane acts as a barrier against environmental stress and makes the membrane permeable; thus, when the intracellular ion extravasation and electrical conductivity were increased, damage to the cell membranes was most severe [44]. Chauhan and Kang found that thymol could destroy the cell membrane of salmonella at 750 µg/mL [45]. A large amount of ROS oxidizes proteins, lipids, carbohydrates, and DNA in living organisms, causing cell membrane rupture or cell death [46,47]. In addition, excessive oxidative stress may lead to a decrease in mitochondrial membrane potential, which in turn leads to an increase in ROS generation [43,48]. Zhang et al. studied the dehydro camelid derivatives, and after treating *R. solani* with the compound I-5, there was membrane damage in the mycelium, so that the intracellular electrolyte seeps into the solution, leading to increased conductivity, the destruction of mitochondrial membrane potential, and, ultimately, to mitochondrial dysfunction, thereby accelerating cell death [49].

Ergosterol, as a unique component of fungal cells, interacts with hydrocarbon bonds on the cell membrane and plays an important role in maintaining membrane fluidity and stabilizing membrane structure [50]. Ergosterol is often used as a target for many antifungal drugs because it only acts in fungal cells [51]. Zhang et al. found that pectin lyase induced the upregulation of ERG4 in cotton Verticillium wilt Vd080, increased the content of ergosterol in the cell membrane, and induced the apoptosis of the pathogen [43]. This is consistent with our results. After the treatment of *R. solani* with lauric acid, the ergosterol content in the mycelium decreased significantly, the synthesis pathway of ergosterol was blocked on the cell membrane, and the cell membrane formed small holes. Consequently, the fluidity and stability decreased. Lauric acid affects the integrity of *R. solani*, through changes in the balance of internal and external osmotic pressure, physiological balance, and so on, which inhibits the mycelium growth of *R. solani*.

Lauric acid can also induce DNA breakage in *R. solani*, that is, it increased the expression of ER molecular chaperones BiP, PDI, and CNX, and downregulated the expression of the ubiquitin-proteasome pathway genes *HSP40*, *HSP70*, and *HSP90*. These changes may lead to misfolded or unfolded proteins (called ER stress) [52], thus inducing the apoptosis of the ER stress cells of *R. solani* (Figure 11) [53,54]. The cell apoptosis of *R. solani* was induced mainly by the ER pathway. Lauric acid can induce ER stress in *R. solani*, improve the level of ER molecular chaperones BiP, PDI, and CNX, promote the increase in Ca^2+^ content in the cytoplasm, and cause DNA breakage and chromatin aggregation, leading to the apoptosis of *R. solani* cells. BiP also binds to misfolded proteins in the ER cavity and acts as ER baroreceptors. BiP can also help fold misfolded proteins or bind them to the ERAD systems [55]. However, if degradation is inadequate, cells will activate multiple apoptotic pathways and induce apoptosis under sustained stress stimulation [43,56]. Lauric acid induced the apoptosis of *R. solani* cells mainly by increasing the level of the ER molecular chaperone protein BiP. This treatment provides a new method for the prevention and treatment of rice sheath blight.

## 5. Conclusions

Through the combined analysis of physiology, biochemistry, and the transcriptome, the preliminary mechanism of action is that lauric acid induces the up-regulation of glucanase and fatty acid metabolism-related genes, reduces the content of ergosterol in cell membranes, and causes cell membrane damage; it induces the down-regulation of cytochrome oxidase gene expression and iron. The up-regulated expression of the reductase gene leads to the accumulation of ROS (H_2_O_2_) in the cell. The overproduction of ROS leads to enhanced cell membrane permeability, electrolyte extravasation, and cell plasma membrane destruction; ROS accumulation also causes DNA fragmentation and ER stress, activating phosphatase, and causing the death of the rice sheath blight pathogen.

## Figures and Tables

**Figure 1 jof-08-00153-f001:**
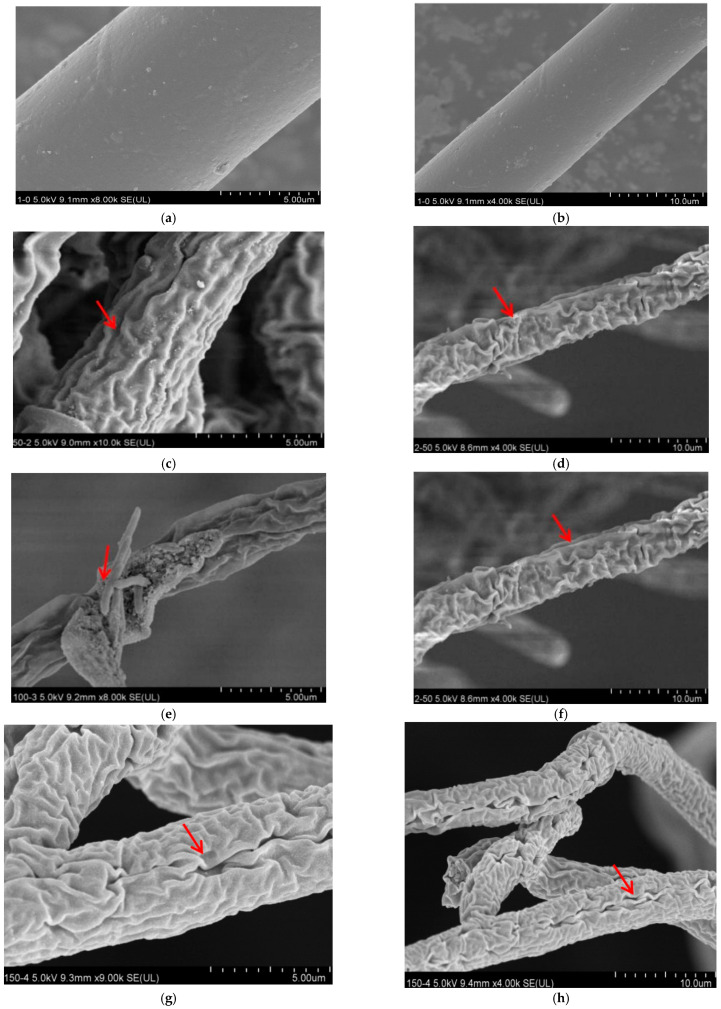
Scanning electron micrographs of *R.*
*solani*. Hyphae exposed to lauric acid at concentrations of (**a**,**b**) 0 µg/mL, (**c**,**d**) 50 µg/mL, (**e**,**f**) 100 µg/mL, (**g**,**h**) 150 µg/mL, and (**i**,**j**) 200 µg/mL. Arrows and arrowheads indicate hyphae shrinkage and partial distortion.

**Figure 2 jof-08-00153-f002:**
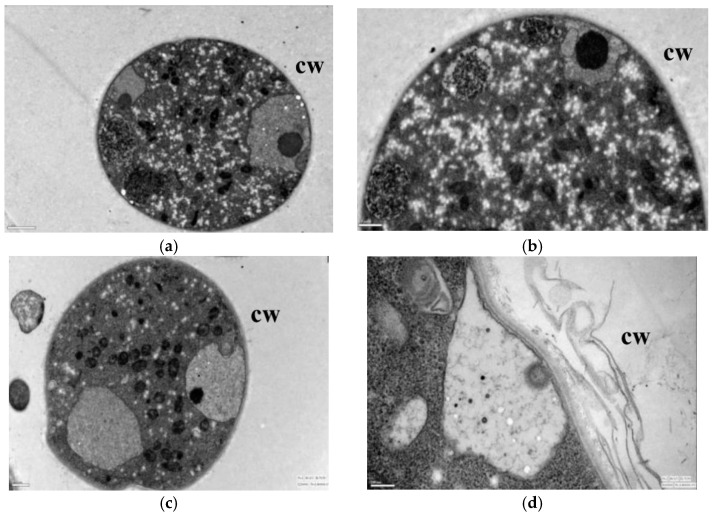
Transmission electron micrographs of *R.*
*solani*. hyphae, where hyphae were exposed to agar with lauric acid at (**a**,**b**) 0 µg/mL, (**c**,**d**) 50 µg/mL, (**e**,**f**) 100 µg/mL, (**g**,**h**) 150 µg/mL, (**i**,**j**) 200 µg/mL. CW = cell wall.

**Figure 3 jof-08-00153-f003:**
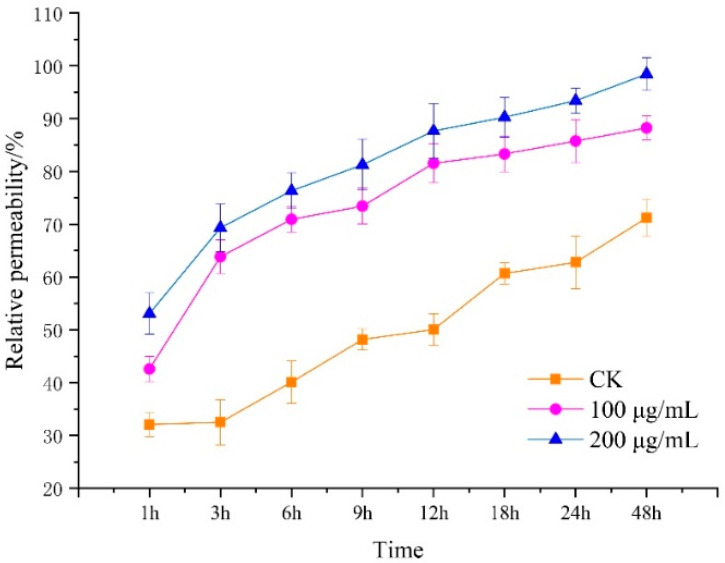
Effect of lauric acid on membrane permeability of the mycelium of *R. solani*.

**Figure 4 jof-08-00153-f004:**
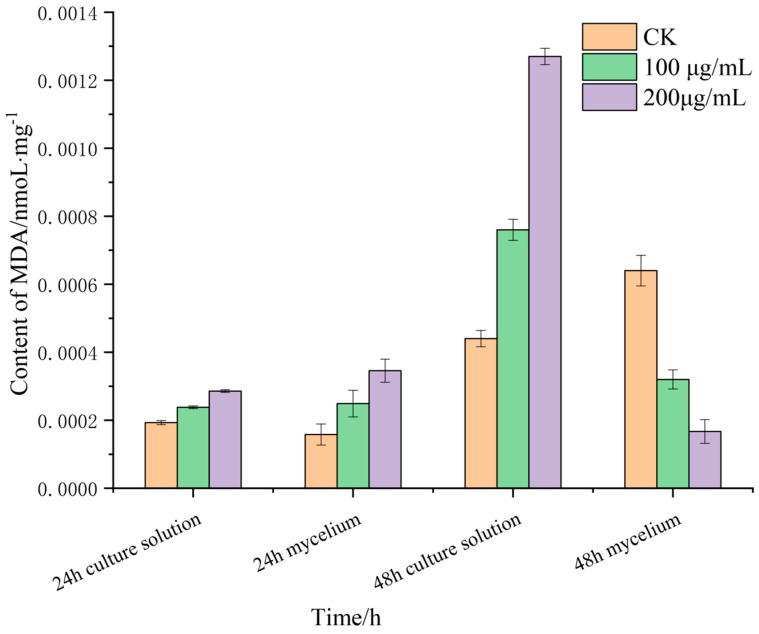
Effect of lauric acid on the MDA content of *R. solani*.

**Figure 5 jof-08-00153-f005:**
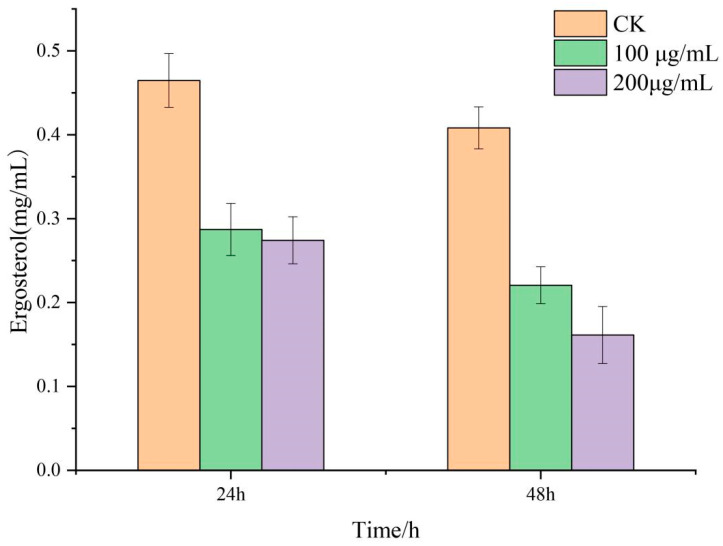
Effect of lauric acid on ergosterol content in hyphae of *R. solani*.

**Figure 6 jof-08-00153-f006:**
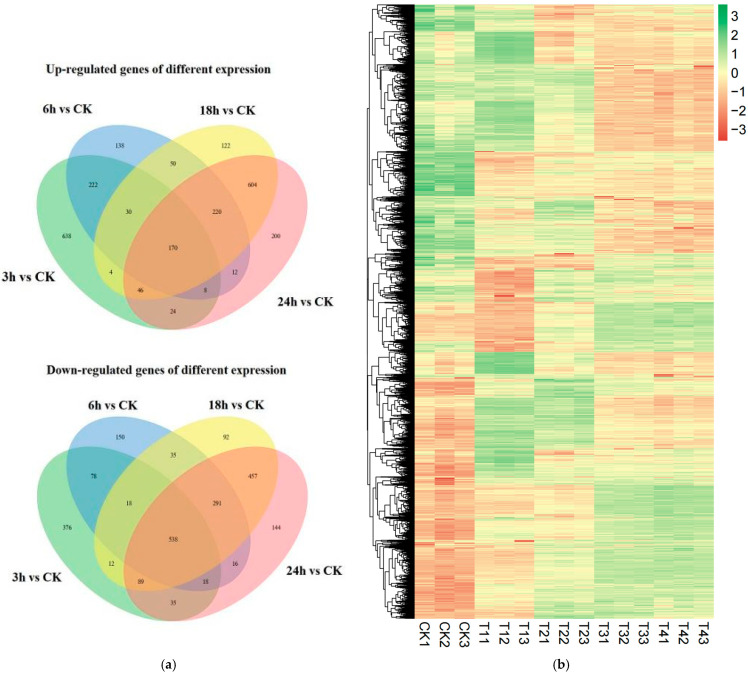
Differentially expressed genes (DEGs) analysis. (**a**) Venn diagram of numbers of the DEGs in different samples. (**b**) Heatmap and cluster analysis of DEGs among different samples using complete linkage. The gene expression level increases with blue to red. CK, samples treated with distill water; 3 h, samples treated with lauric acid for 3 h; 6 h, samples treated with lauric acid for 6 h; 18 h, samples treated with lauric acid for 18 h; 24 h, samples treated with lauric acid for 24 h.

**Figure 7 jof-08-00153-f007:**
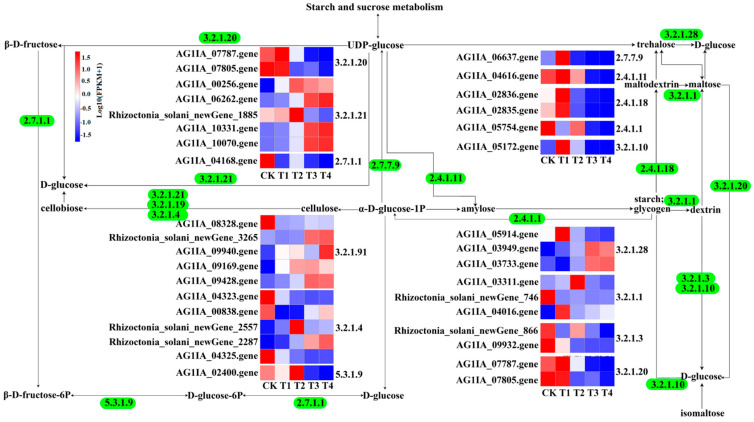
Selected genes related to starch and sucrose metabolism in *R. solani* under four treatments. Bounding boxes represent metabolic processes or metabolite, and digits represent regulatory enzyme for a specific process. Straight arrows indicate the transformational direction of a metabolite (solid and dotted lines correspond to direct and indirect effects). The colors indicate the level of each gene’s expression based on FPKM normalized read counts.

**Figure 8 jof-08-00153-f008:**
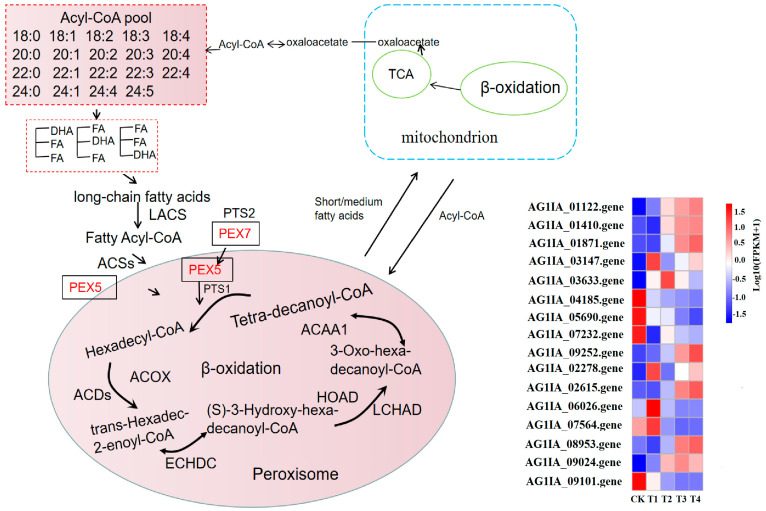
Effects of lauric acid on fatty acid metabolism of rice sheath blight.

**Figure 9 jof-08-00153-f009:**
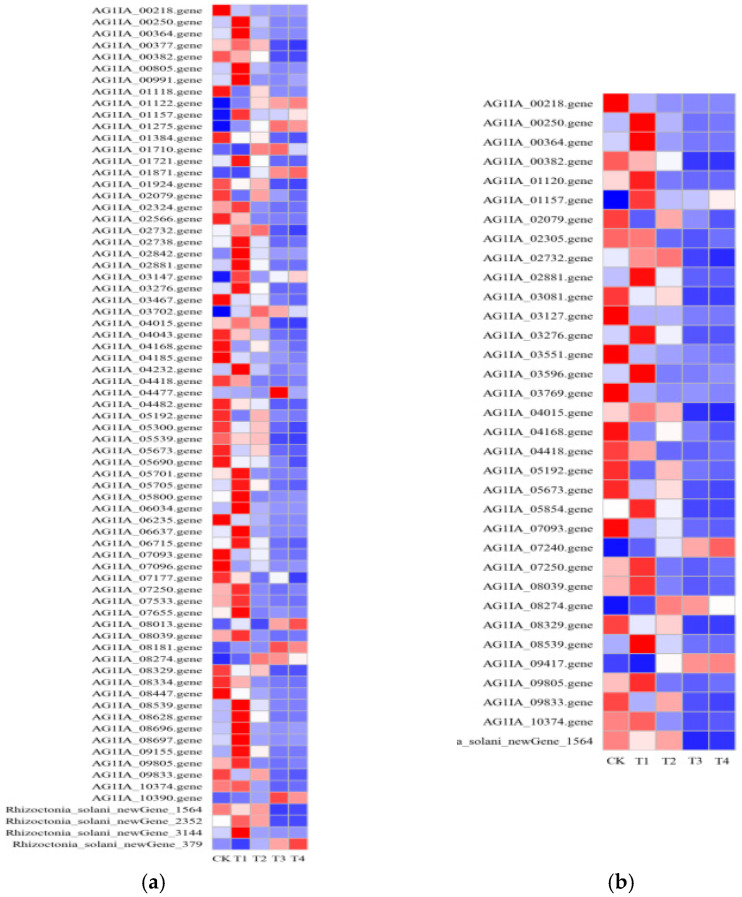
Effects of lauric acid on biosynthesis of antibiotics and carbon metabolism of rice sheath blight; (**a**) Antibiotic biosynthesis-related genes; (**b**) Carbon metabolism synthetic-related genes.

**Figure 10 jof-08-00153-f010:**
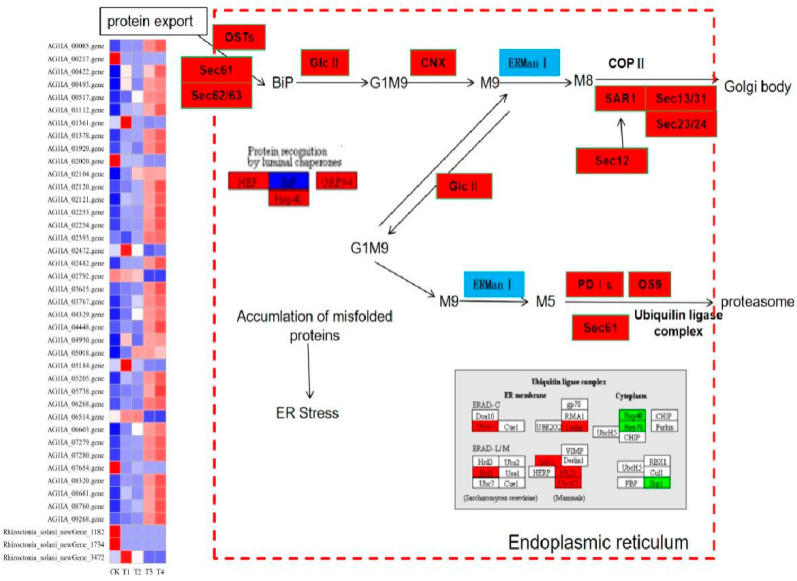
Analysis of protein processing in endoplasmic reticulum pathways in lauric acid-induced rice sheath blight.

**Figure 11 jof-08-00153-f011:**
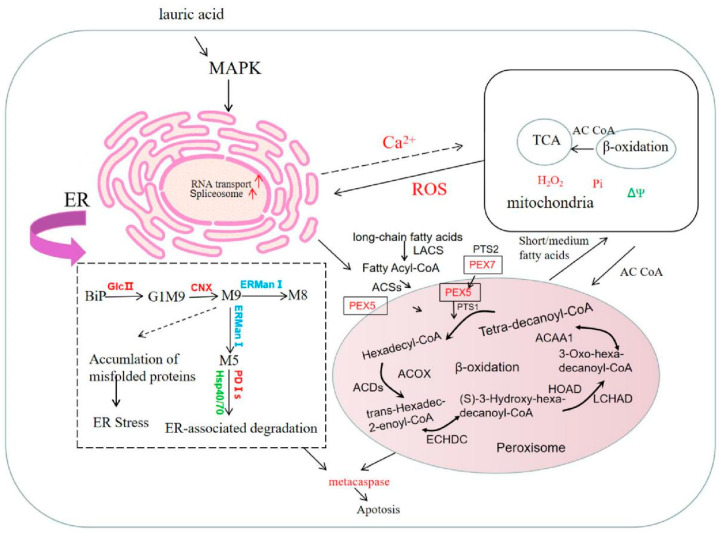
Mechanism model of cell apoptosis induced by lauric acid.

**Table 1 jof-08-00153-t001:** Minimum inhibitory concentration of lauric acid on rice sheath blight.

Lauric Acid Treatment (µg/mL)	1d	2d	3d
0	++	+++	++++
50	++	++	++
100	+	+	+
150	−	+	+
200	−	−	+
250	−	−	+
300	−	−	−
350	−	−	−

Note: “−”: no growing; “+”: growing.

**Table 2 jof-08-00153-t002:** Summary of the RNA-Seq data analysis.

Samples	Total Reads	Clean Reads	Clean Bases	GC Content (%)	% ≥Q30 (%)	Total Mapped
CK1	64,784,226	32,392,113	9,678,905,954	53.27	94.59	58,992,337(91.06)
CK2	61,435,340	30,717,670	9,186,324,764	53.33	94.28	56,073,649(91.27)
CK3	64,967,078	32,483,539	9,712,848,232	53.25	94.7	59,354,850(91.36)
3h-1	65,541,784	32,770,892	9,803,952,030	53.11	94.28	59,711,907(91.11)
3h-2	69,943,128	34,971,564	10,457,950,356	53.12	94.42	63,761,674(91.16)
3h-3	60,559,660	30,279,830	9,059,999,778	53.13	94.48	55,045,427(90.89)
6h-1	62,595,230	31,297,615	9,357,641,672	53.23	94.37	56,786,050(90.72)
6h-2	58,713,586	29,356,793	8,781,141,820	53.22	94.34	53,231,872(90.66)
6h-3	63,140,638	31,570,319	9,437,986,814	53.21	94.05	57,316,763(90.78)
18h-1	68,915,458	34,457,729	10,297,492,078	53.02	94.29	62,304,463(90.41)
18h-2	72,425,996	36,212,998	10,835,409,898	53.14	94.41	65,386,591(90.28)
18h-3	72,620,234	36,310,117	10,851,217,652	53.13	94.15	65,585,306(90.31)
24h-1	67,231,244	33,615,622	10,054,713,440	53.11	94.07	60,647,299(90.21)
24h-2	71,866,228	35,933,114	10,750,769,100	53.14	94.09	64,791,542(90.16)

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
