# Peer review of "Lauric Acid Induces Apoptosis of Rice Sheath Blight Disease Caused by Rhizoctonia solani by Affecting Fungal Fatty Acid Metabolism and Destroying the Dynamic Equilibrium of Reactive Oxygen Species"

_jof, 2022, doi:10.3390/jof8020153_

Round 1
Reviewer 1 Report
The paper “Lauric acid induces apoptosis of rice sheath blight disease caused by Rhizoctonia solani by affecting fungal fatty acid metabolism and destroying the dynamic equilibrium of reactive oxygen species” written by Wang et al., has analyzed the molecular mechanism of Lauric acid effects in R. solani through the transcriptome studies. Their study has indicated the potential use of lauric acid could as control of rice sheath blight. A comprehensive analysis has been done, but the manuscript needs revision for any grammatical mistakes and unexplained procedures. I have some following points for the change in manuscript…
1. Many scientific names of organisms are not italic, recheck throughout the manuscript.
2. “Richard et al. reported that the compound film of lauric acid (LDPE) can inhibit Colletotrichum tamarilloi, R. solani, and Pythium ultimum of wood tomato and that it is difficult for these bacteria to develop resistance to lauric acid[21, 22]” : Check the references and rewrite.
3. “while du-ensiform gall showed a good toxicological effect”: Is not clear and not supported by the reference. Need the support of reference.
4. In the material section, storage is mentioned at -80. Mention the normal storage also. Mention the purpose of storing at -80
5. Is it “Institute of Crop Protection” or Crop protection institute? The redundancy related to the strain in the materials and method section needs to remove.
6. “tested lauric acid was diluted” what does it mean?
7. Line 109-113 need revision for clarity,
8. MIC full form is not mentioned
9. Incomplete sentence “Details of the wang et al.[50]treatment.”
10. Consistency is not mentioned in writing the genus name of Rhizoctonia. It needs to mention as “R.”.
11. 123-129 need revision for clarity of steps followed for conducting the experiment.
12. Line 132-139 needs revision for grammar. Also, the standard curve preparation is not mentioned.
13. How was the “grinding the hyphae at 60℃” achieved?
14. Line 144 : How much water and chloroform is added?
15. “ot was dried on a filter”, what is OT?
16. The whole method section needs revision for clarity and steps should be explained properly.
17. Line 205: check units
18. Please check grammatical mistakes in the results and discussion.
Author Response
Point 1: Many scientific names of organisms are not italic, recheck throughout the manuscript.
Response 1: Thank you very much for your review of my manuscript and your hard work. I have carefully confirmed the naming of species and genera in the article, and there are indeed many errors. Thank you very much for your correction opinions. I have carefully revised and standardized the naming of species in the article according to your requirements, And the italicized writing part is italicized.
Point 2: “Richard et al. reported that the compound film of lauric acid (LDPE) can inhibit Colletotrichum tamarilloi, R. solani, and Pythium ultimum of wood tomato and that it is difficult for these bacteria to develop resistance to lauric acid[21, 22]” : Check the references and rewrite.
Response 2: Thank you for your correction of my article. I read the full text of references [21, 22] carefully and found that I had misunderstandings in this part. Thank you very much for your correction of my error. I have revised this part to ’Richard et al. reported that the compound film of lauric acid (LDPE) can inhibit Colletotrichum tamarilloi, R. solani, and Pythium ultimum of wood tomato, so lauric acid could play an antifungal role in vivo and in vitro[21, 22].
Point 3: “while du-ensiform gall showed a good toxicological effect”: Is not clear and not supported by the reference. Need the support of reference.
Response 3:Thank you very much for pointing out the part of my article that makes readers unable to accurately understand or confuse you. I want to say sorry for the difficulty or confusion caused by the article. This part is a part of my early experimental results and has been published.They can kill or inhibit the growth of target pathogens efficiently with no or low toxicity, degrade readily, do not prompt development of resistance, which has led to their widespread use. In this study, the growth inhibition effect of 24 plant-sourced ethanol extracts on rice sprigs was studied. Ethanol extract of gallnuts and cloves inhibited the growth of rice sprites by up to 100%. Indoor toxicity measurement results showed that the gallnut and glove constituents inhibition reached 39.23 μg/mL and 18.82 μg/mL, respectively. Extract treated rice sprigs were dry and wrinkled. Gallnut caused intracellular swelling and breakage of mitochondria , disintegration of nuclei, aggregation of protoplasts, and complete degradation of organelles in hyphae and aggregation of cellular contents. Protection of Rhizoctonia solani viability reached 46.8% for gallnut and 37.88% for clove in water emulsions of 1,000 μg/mL gallnut and clove in the presence of 0.1% Tween 80. The protection by gallnut was significantly stronger than that of clove. I have provided references, which can eliminate this part of confusion for readers so that they can understand it more clearly.
Point 4:In the material section, storage is mentioned at -80. Mention the normal storage also. Mention the purpose of storing at -80
Response 4:Thank you very much for your good suggestions. In this part, I propose that the purpose of - 80 ℃ preservation is to preserve the identified strains, so that they can be preserved longer, make the future experiments more convenient and ensure the consistency of strains,The specific operation of strain preservation at 80 ℃ is: inoculate Rice Sheath Blight in a liquid PDA triangular flask to make its mycelial growth concentration OD600 above 0.6, mix the bacterial solution with 40% glycerol (1:1), and then store the rice sheath blight preserved in a glycerol tube in an ultra-low temperature ice box at - 80 ℃.
Point 5:Is it “Institute of Crop Protection” or Crop protection institute? The redundancy related to the strain in the materials and method section needs to remove.
Response 5:Thank you very much for your question. I have carefully read and revised this part of your content. This part is indeed inaccurate in my description. I have revised this part, revised it according to your requirements, and deleted the part in the method. Finally, this part is uniformly revised to Rhizoctonia solani was sampled in Changsha, Hunan Province. It was identified as a strong pathogenic strain (R. solani AG1IA) by the Institute of Crop Protection of Guizhou University. R. solani cultures was were maintained on potato dextrose agar (PDA) at 28 °C and 150 rpm for 3 d. Some bacterial liquid was mixed with 40% glycerol (v:v=1:1) and stored at - 80 ℃.
Point 6:“tested lauric acid was diluted” what does it mean?
Response 6:Thank you very much for your questions on this part. I'm sorry for the confusion caused by this part. This sentence mainly means diluting lauric acid into different concentrations to further study the inhibitory effect and minimum inhibitory concentration of lauric acid on rice sheath blight. This part mainly explains that it means that PDA was poured into sterilized Petri dishes (9 cm diameter) and lauric acid was added to PDA culture mediums to make the concentration reach 0, 50, 100, 150 and 200 µg/mL,Nonstandard expression is your confusion in understanding. I have revised it to make it clearer. Finally, this part is revised to ’The antifungal tests of lauric acid was carried out for assessing the effects towards mycelial growth of R. solani as described previously[26]. For determination of contact effects, lauric acid was dispersed as an emulsion in water using 5%OP-15(Beijing Solarbio Science&Technology Co., LTD) (10% v/v) and added to PDA immediately before it was emptied into the glass Petri dishes (90 mm in diameter) at a temperature of 35-40°C.The concentrations tested were 50 to 350 μg/mL.
Point 7:Line 109-113 need revision for clarity
Response 7:Thank you very much for your hard work on my article. I will carefully revise the questions you raised, because it makes the article easier to understand, and it also makes the language description of the article more standardized and clear. This part is mainly because I can't describe clearly in the process of narration, which leads to confusion in understanding. I'm sorry to bring confusion to your reading. I did a careful reading of this part and made a careful revision to make it more clear. I have revised this part to the antifungal tests of lauric acid was carried out for assessing the effects towards mycelial growth of R. solani as described previously[26]. For determination of contact effects, lauric acid was dispersed as an emulsion in water using 5%OP-15(Beijing Solarbio Science&Technology Co., LTD) (10% v/v) and added to PDA immediately before it was emptied into the glass Petri dishes (90 mm in diameter) at a temperature of 35-40°C.The concentrations tested were 50 to 350 μg/mL. The controls received the same quantity of 5%OP-15 mixed with PDA. After solidification, a 5 mm diameter disc cut from the actively growing front of a 3-day old colony of the desired pathogenic fungus was then placed with the inoculum side down in the center of each treatment plate, aseptically. Treated petri dishes were then incubated at 28°C till the fungal growth was almost complete in the control plates. All experiments were in quadruplet for each treatment against R. solani. To determine the MIC and lethal dose (LD) of lauric acid, respectively, the growth of hyphae of R. solani on each plate was observed after 1, 2, and 3 d of culture. The formula for calculating the growth inhibition of fungal hyphae was as follows:
Inhibition(%)= (1-Dt/Dc)×100
where Dt and Dc were the growth zone diameters in the experimental dish (mm) and the control dish (mm), respectively.Every treatment has three replicates. Based on previous studies, the EC50 value was obtained by regressing growth inhibition rate against the log of the lauric acid concentration. Every treatment has three replicates.
Point 8:MIC full form is not mentioned
Response 8:Thank you very much for your question. I have carefully read the MIC part of the article. MIC is mainly the screening process of the minimum concentration of lauric acid inhibiting rice sheath blight. It is the basis of the following experiments. Through the correlation between lauric acid concentration and bacterial production, we can judge the minimum inhibitory concentration of lauric acid on rice sheath blight. This part of the article focuses on the experiments of lauric acid mic and EC50.In the minimum inhibitory concentration (MIC) test of lauric acid, different concentrations of lauric acid showed different antibacterial effects after 2 d of culture (Table 1). Furthermore, this effect exhibited a dose-dependent effect. When the lauric acid concentration of lauric acid was less than 100 µg/mL, the plate was full of hyphae. When the lauric acid concentration was 50–200 µg/mL, a small amount of hyphae appeared on the plate. When the lauric acid concentration was greater than or equal to 250 µg/mL, no hyphae appeared on the plate. Therefore, the lowest sensitive concentration of lauric acid for inhibition of mycelium growth was determined to be 150 µg/mL,Finally, the MIC of lauric acid was 150 µg/mL.
Point 9:Incomplete sentence “Details of the wang et al.[50]treatment.”
Response 9:Thank you very much for your review of the article. Your seriousness has made me learn more and benefited me a lot in writing. Thank you very much! I carefully analyzed the content of this part and found that the description was incomplete and did not express the real content. I carefully revised this part and revised it to PDA was poured into sterilized Petri dishes (9 cm diameter) and lauric acid was added to PDA culture mediums to make the concentration reach 0, 50, 100, 150 and 200 µg/mL. Culture plates were incubated at 28°C for 3 d in darkness. The hyphae were used for Scanning electron microscopy(SEM) and transmission electron microscopy(TEM) observations. Rectangular blocks (0.5 cm × 0.3 cm) from the edge of the mycelium were placed in a centrifuge tube with 1 mL of 25% dialdehyde fixation fluid. Three blocks were taken for each treatment. Each sample was suctioned repeatedly with a 50 mL syringe until the bubbles on the surface of the mycelium disappeared. The centrifuge tube was sealed and stored overnight at 4°C. After suction, the retaining fluid was carefully rinsed three times with 0.1 M PBS for 10 min each time. Then, 0.5 mL of 1% nitric acid fixative was added within 2 h. Each sample was washed three times with PBS. Ethanol solutions of 30%, 50%, 70%, 80%, and 90% were used for dehydration for 10 min each time, followed by dehydration twice with waterless ethanol for 10 min each time. After dehydration, the specimens were dried in a freeze drier (LGJ-10D; Beijing Fourth Ring Scientific Instrument Co., Ltd., Beijing, China), and sputter-coated with gold. Microscopy was performed using a SEM (S-3400N; Hitachi,Tokyo, Japan) operated at an accelerating voltage of 20 kV. Controls consisted of untreated mycelia, which were prepared in parallel with experimental samples.
The mycelium collected as described above was poured into a centrifuge tube with 1 mL of 2.5% dialdehyde fixation fluid. The tube was sealed and incubated overnight at 4°C. After the remaining fluid was carefully suctioned off, the sample was rinsed three times with 0.1 M PBS, for 10 min each time. Then, 0.5 mL of 1% nitric acid fixative was added within 2 h. They were then washed three times with PBS, and ethanol solutions with concentrations of 30%, 50%, 70%, 80%, and 90% were dehydrated for 10 min each time, and then dehydrated twice with waterless ethanol for 20 min each time. A mixture of acetone and resin with different concentration ratios (3:1, 1:1, 1:3 v/v ) was successively used for infiltration for 3 h each time. This was followed by treatment with pure resin overnight. After polymerization at 70°C for 24 h, the embedded samples were removed for preparation of ultra-thin sections. The sections were stained with lead citrate and uranium diacetate, dried, and observed by TEM.’
Point 10:Consistency is not mentioned in writing the genus name of Rhizoctonia. It needs to mention as “R.”.
Response 10:Thank you very much for your review of the article. I read the article carefully and found that it is indeed not explained clearly in the article. When abbreviating, first explain it when it appears for the first time, and then abbreviate it below, so as not to bring confusion to reading. I have carefully revised it according to your requirements and explained it when it first appeared in the text. 'R.' is the abbreviation of 'Rhizoctonia'.
Point 11:123-129 need revision for clarity of steps followed for conducting the experiment.
Response 11:Thank you very much for asking good questions. In order to better understand and apply for readers, the experimental methods should be more specific in order to better verify the experimental results and make readers better understand the specific content of this paper, so as to more comprehensively understand and apply the experimental results. In this way, the content of the article is richer, and the experimental method is also one of the effective ways to obtain the results. We can better understand which experimental method should be selected to solve this series of problems, which is more targeted and repeat the experimental results, so as to provide reference for readers. I have further refined and described the experimental steps clearly. I change it to
- solani cake (5 mm) was taken and added to 100 mL of potato dextrose broth (PDB) medium, and kept in a 28℃ shaker (150 r/h) for 24 h to form a uniform hyphal suspension. Lauric acid incubated R. solani mycelia were gathered and washed twice for 2 min with sterilized water. One gram of the mycelia was placed in 15 mL centrifuge tubes containing one of three concentrations of lauric acid(0, 100, and 200 μg/mL). The background value of electrical conductivity was measured by a DDS-307 conductivity meter (J0); then after 0 h, 3 h, 6 h, 9 h, 12 h, 24 h, 48 h, according to the methods of Li et al.[27]with some modifications, 10 mL of the mixture was taken and centrifuged (4,000 rpm; 5 min). Its conductivity was measured, with supernatant J1. Then, the mixture was boiled for 15 min, cooled, and centrifuged. The conductivity measured in the supernatant was recorded as J2. Finally, the relative penetration of each time period was calculated. Permeabilities (P%) were calculated by the following formula: P%=[(J1-J0)/(J2-J0)].
Point 12:Line 132-139 needs revision for grammar. Also, the standard curve preparation is not mentioned.
Response 12:Thank you very much for your hard work and your questions. I carefully read the problems existing in this part of the content and find corresponding professionals to standardize and modify the language to correct the existing grammatical errors. This part is revised to Thiobarbituric acid method was used to determine the malondialdehyde(MDA) activity of lauric acid. One gram of the mycelium was weighed, 2 mL of 10% trichloroacetic acid (TCA) and a small amount of quartz sand were added. This was ground to a homogenate, 8 mL of TCA was added, and the homogenate was centrifuged at 4,000 r/min for 15 min, leaving the supernatant for later use. Two milliliters of the supernatant from centrifugation (adding 2 mL distilled water to the control) was taken and added with 2 mL of 0.6% thiobarbituric acid solution. Then, the mixture was allowed to react in a boiling water bath for 15 min, then cooled down to room temperature. Absorbance was measured at 532 nm, 600 nm,and 450 nm (Beckman Coulter DU800 (Daniel L. Arnon), and the content was determined from a standard curve and calculated as the following formula: MDA(mmol·g-1FW)=[6.45×(D532-D600)-0.56×D450]×[Vt/(Vs×W)]
Vt:Total volume of extract(mL); Vs:Volume of extract for determination(mL); FW:Fresh weight of extracted tissue(g). We have further improved the standard curve you raised and further explained the production of the standard curve. Our standard curve is not included in the article, but a brief description of the production process of the standard curve. We have explained in detail the steps of making the standard curve. We amend this part to Ergosterol standard is dissolved in methanol to make stock solution and diluted into ergosterol standard (the concentration of ergosterol: 0.016, 0.032, 0.064, 0.80, 0.120 μ g/mL). Carry out linear regression according to the peak area and the corresponding mass concentration, and draw the standard curve.
Point 13:How was the “grinding the hyphae at 60℃” achieved?
Response 13:Thank you very much for your review of the article. I read this part carefully. This part is due to errors in the narrative process. It originally meant that the culture medium of rice sheath blight treated with lauric acid was collected and removed, dried at 60 ℃ and ground the sample after drying. Please forgive me for the inconvenience caused by the ambiguity in the narration. I have revised the content of the article to make it clearer. The article is revised as the extraction method of ergosterol from R. solani included collecting the treated hyphae, ,grinding the hyphae after drying at 60℃.
Point 14:Line 144 : How much water and chloroform is added?
Response 14:Thank you very much for your valuable comments on the article, so that there are fewer mistakes in the article. You proposed how much water and chloroform should be added when extracting ergosterol. I carefully read this part of the article. In the article, when extracting ergosterol from rice sheath blight treated with lauric acid, add 10ml of water, 10ml of chloroform and 10ml of 0.5 mol / L phosphate buffer containing 2.0 mol / L KCl for extraction. The amount of these three kinds of addition is 10ml, that is, water: chloroform: 0.5 mol / LPBs (containing 2.0 mol / L KCl) = 1:1:1. Please forgive me for the trouble in reading the article. I have revised it to this .”On the next day, water, chloroform, and 0.5 mol/L phosphate buffer containing 2.0 mol/L KCl were added in turn each 10.0 mL after mixing ,and extracting, and the chloroform phase was dried by nitrogen blower at 60℃.”
Point 15:“ot was dried on a filter”, what is OT?
Response 15:Thank you very much for your hard work on this article. At the same time, thank you for reading it carefully. I read the article carefully and found that there were spelling mistakes in this part of the content. I have changed 'ot' to 'it'.
Point 16:The whole method section needs revision for clarity and steps should be explained properly.
Response 16:Thank you very much for your valuable comments on the article, and thank you very much for reading the article carefully. I have carefully read and revised some contents of the method, and revised the existing problems one by one according to your requirements. The revised details are marked in the article, and the contents are standardized and modified by professionals.
Point 17:Line 205: check units
Response 17:Thank you very much for your hard work in this article. I read the part you proposed carefully and found that there were unit spelling errors in this part. We carefully revised this part and changed 'g / mL' to '' μ g/mL’.
Point 18:Please check grammatical mistakes in the results and discussion.
Response 18:Thank you very much for your hard work and valuable comments on the article. I carefully read some of the contents of your comments, carefully revised the existing problems, and revised the grammatical errors in the article. Through the standardized revision by professionals, the article is easier to understand and understand, and the problems are more simple and clear. Thank you very much for your review of the article. I will carefully revise and revise the questions raised in the article. Please forgive me for the trouble brought to you in the article. I will further correct the grammar and spelling mistakes in time, and further supplement and standardize the language in the experimental steps in the article.
At last, I sincerely want to say happy new year and I hope you have a nice day!

Reviewer 2 Report
The manuscript shows the effect of Lauric acid in induction of apoptosis in Rhizoctonia solani by affecting fungal fatty acid metabolism and destroying the dynamic equilibrium of reactive oxygen species. Although the methods for this investigations well programmed but can not confirm that the Lauric acid would able to control Rice sheath blight disease. To confirm that the lauric acid is able to control sheath blight disease the authors need to do some invivo experiments on rice plant. Therefore, I suppose to the authors that delete all statements which are about disease control. In this manuscript the authors investigated the effects of lauric acid on Rhizoctonia solani and they did not do any experiment on disease control of rice.
Author Response
Point 1:The manuscript shows the effect of Lauric acid in induction of apoptosis in Rhizoctonia solani by affecting fungal fatty acid metabolism and destroying the dynamic equilibrium of reactive oxygen species. Although the methods for this investigations well programmed but can not confirm that the Lauric acid would able to control Rice sheath blight disease. To confirm that the lauric acid is able to control sheath blight disease the authors need to do some invivo experiments on rice plant. Therefore, I suppose to the authors that delete all statements which are about disease control. In this manuscript the authors investigated the effects of lauric acid on Rhizoctonia solani and they did not do any experiment on disease control of rice.
Response 1: Thank you very much for your meaningful question, which is why lauric acid is the ultimate goal of controlling rice sheath blight. It is also the starting point for us to apply our results to the control of rice sheath blight. We also considered whether to include this part of your question in this manuscript. In the end, we decided to leave this part out of the manuscript so that we could better explain the mechanism of action of lauric acid against rice sheath blight at the physiological, biochemical and molecular levels. Our experiments on the control of Rice Sheath blight have also been completed. The results of indoor and field experiments show that lauric acid has a certain control effect on rice sheath blight, and the lauric acid expected to be further developed as a new pesticide, therefore, the control mechanism of lauric acid was studied.
1.Protection effects of lauric acid on rice plants
The control effect of 200-1000 µg/mL lauric acid on rice sheath blight was as shown in Table 1. With the concentration of 200 µg/mL lauric acid, the control effect was 18.11% and 1000 µg/mL was 75.34%.The results showed that the differences were very significant and significant at 1% and 5% level.
Table 1. Indoor control effect of lauric acid on rice sheath blight
Treatment |
Disease Index |
Control effect (%) |
Average control effect(%) |
||
â… |
â…¡ |
â…¢ |
|||
CK |
5.19 |
|
|
|
|
200 µg/mL |
4.25 |
18.50 |
19.27 |
16.57 |
18.11eE |
400 µg/mL |
2.92 |
44.89 |
43.55 |
42.97 |
43.80 dD |
600 µg/mL |
2.16 |
58.00 |
58.38 |
58.96 |
58.45cC |
800 µg/mL |
1.55 |
70.91 |
69.94 |
69.75 |
70.20bB |
1000 µg/mL |
1.28 |
74.76 |
75.72 |
75.53 |
75.34 aA |
Note: the same column data is not contain the same letter mean to the difference is significant at 1% and 5% levels
Figure 1. Indoor control effect of lauric acid on rice sheath blight
2. Field control effect of lauric acid film on Rice Sheath Blight
The field control effect of lauric acid film on rice sheath blight was shown in Table 2. The field control effect of 800 g/667m2 of lauric acid film after 7 days was 93.19% , which was higher than that of Jinggangmycin, and the difference was significant at 5% level. The control effect of 50g/667m2 of lauric acid was 61.72% after 7 days, which was slightly lower than that of Jinggangmycin, and there was difference at 5% level.
Table 2. Control effect of 22.1% galla chinensis film on rice sheath blight
Treatment |
Dosage of preparation(g/667 m2) |
7 d |
14 d |
||
DI(%) |
Control effect (%) |
DI(%) |
Control effect (%) |
||
15% Jinggangmycin WP |
50 |
2.28 |
64.04e |
5.96 |
24.71f |
lauric acid film |
800 |
0.43 |
93.19a |
0.98 |
89.49a |
lauric acid film |
400 |
0.98 |
88.76b |
1.46 |
80.37b |
lauric acid film |
200 |
1.68 |
78.09c |
2.59 |
70.66c |
lauric acid film |
100 |
1.23 |
75.80d |
2.96 |
67.88d |
lauric acid film |
50 |
2.44 |
61.72f |
3.69 |
61.61e |
At last, I sincerely want to say happy new year and I hope you have a nice day!
